# Adsorption and Photocatalytic Degradation of Pesticides into Nanocomposites: A Review

**DOI:** 10.3390/molecules27196261

**Published:** 2022-09-23

**Authors:** Franciele S. Bruckmann, Carlos Schnorr, Leandro R. Oviedo, Salah Knani, Luis F. O. Silva, William L. Silva, Guilherme L. Dotto, Cristiano R. Bohn Rhoden

**Affiliations:** 1Laboratório de Materiais Magnéticos Nanoestruturados, LaMMaN, Universidade Franciscana-UFN, Santa Maria 97010-032, RS, Brazil; 2Programa de Pós-Graduação em Nanociências, Universidade Franciscana-UFN, Santa Maria 97010-032, RS, Brazil; 3Department of Civil and Environmental, Universidad de la Costa, CUC, Calle 58 # 55–66, Barranquilla 080002, Atlántico, Colombia; 4College of Science, Northern Border University, Arar 91431, Saudi Arabia; 5Laboratory of Quantum and Statistical Physics, Faculty of Sciences of Monastir, University of Monastir, Monastir 5079, Tunisia; 6Research Group on Adsorptive and Catalytic Process Engineering (ENGEPAC), Department of Chemical Enginnering, Federal University of Santa Maria, Santa Maria 97105-900, RS, Brazil

**Keywords:** emerging pollutants, nanotechnology, sustainability

## Abstract

The extensive use of pesticides in agriculture has significantly impacted the environment and human health, as these pollutants are inadequately disposed of into water bodies. In addition, pesticides can cause adverse effects on humans and aquatic animals due to their incomplete removal from the aqueous medium by conventional wastewater treatments. Therefore, processes such as heterogeneous photocatalysis and adsorption by nanocomposites have received special attention in the scientific community due to their unique properties and ability to degrade and remove several organic pollutants, including pesticides. This report reviews the use of nanocomposites in pesticide adsorption and photocatalytic degradation from aqueous solutions. A bibliographic search was performed using the ScienceDirect, American Chemical Society (ACS), and Royal Society of Chemistry (RSC) indexes, using Boolean logic and the following descriptors: “pesticide degradation” AND “photocatalysis” AND “nanocomposites”; “nanocomposites” AND “pesticides” AND “adsorption”. The search was limited to research article documents in the last ten years (from January 2012 to June 2022). The results made it possible to verify that the most dangerous pesticides are not the most commonly degraded/removed from wastewater. At the same time, the potential of the supported nanocatalysts and nanoadsorbents in the decontamination of wastewater-containing pesticides is confirmed once they present reduced bandgap energy, which occurs over a wide range of wavelengths. Moreover, due to the great affinity of the supported nanocatalysts with pesticides, better charge separation, high removal, and degradation values are reported for these organic compounds. Thus, the class of the nanocomposites investigated in this work, magnetic or not, can be characterized as suitable nanomaterials with potential and unique properties useful in heterogeneous photocatalysts and the adsorption of pesticides.

## 1. Introduction

The intense populational growth and industrial expansion in the most diverse segments of society have led to a substantial increase in the demand for drinking water supply and large-scale food production [1]. Thus, to increase productivity at an economically profitable level, the employment of agrochemicals has been widely used to combat pests and weeds [2]. However, pesticides are characterized by low biodegradability, high bioaccumulative capacity arising from their physicochemical properties, and a long half-life, of 5–15 years, increasing their toxicity to the environment and humans [3,4]. Thus, pesticide persistence in soil, wastewater, ground, and surface water has proved to be a considerable environmental problem, and may be compounded along the food chain, reaching concentrations toxic to human health [5]. Due to their high stability, these compounds can contaminate areas distant from pulverization through water volatilization and soil absorption. Studies have associated exposure to compounds with hormonal changes in the immune, neurological, and cardiac systems, as well as with the development of neoplasms [6,7].

For this purpose, diverse techniques, such as adsorption and advanced oxidative processes (AOPs), which include Fenton, photo-Fenton, heterogeneous photocatalysis, and ozonation systems, have been explored for removing and degrading biopersistent organic compounds [8,9,10]. AOPs are based on the generation of free radicals, e.g., hydroxyl (HO•) and superoxide (O_2_^−^•) radicals, which have high oxidizing power in an aqueous solution and are able to degrade pollutants into lower molecular weight intermediates and inorganic precursors [11,12]. Heterogeneous photocatalysis is an advanced oxidative process that occurs through the photoactivation (by sunlight or artificial light) of a semiconductor, which uses water molecules and dissolved oxygen as reagents of oxi-reduction reactions [13]. This technique is very efficient and promising for the degradation of organic pollutants, including dyes, drugs, and pesticides [10,14,15]. Among the materials used, metallic nanooxides (zinc oxide and titanium dioxide) have been largely employed due to their excellent properties, such as low toxicity, good availability, chemical stability, large surface area/porosity, and photocorrosion [16,17]. However, these conventional nanocatalysts are characterized by their high bandgap energy, which is the energy required to start photocatalytic reactions. Additionally, due to their high surface energy, they tend to agglomerate during the photocatalytic process. Therefore, the association of these nanocatalysts with a second, less active material (called catalytic support or matrix) can solve these drawbacks, even when the active material is dispersed in low concentrations (ca. 0.5–5 wt%) on the support [18]. Thus, combining the two materials results in a new material called nanocomposite, in which the active substance is in the above-mentioned concentration range and this is named the reinforcement phase [19].

Nanocomposites are multiphase materials formed by a continuous and dispersed phase and have at least one dimension in the nanoscale [20]. The continuous phase (matrix) consists of a compound of polymeric, ceramic, or metallic origin, while the dispersed phase (reinforcement) is commonly derived from fibrous materials [21,22,23]. Nanocomposite materials are synthesized to combine individual properties and reduce limitations, such as physicochemical and thermal instability, expanding the scope of applications [22]. In parallel, at the nanoscale, the materials exhibit distinct behaviors to those found at the micrometer scale, such as volume/area relationship and increased reactivity [24].

Another technique widely used for pesticide removal from wastewater consists of adsorption, especially when using nanomaterials (adsorbents), due to its simplicity of operation, relatively low cost, and low energy requirements [25]. In addition, nanoadsorbents are characterized by their high specific surface area, chemical/thermal stability, and affinity for organic pollutants [26]. Although the efficiency of nanoadsorbents in the removal of organic compounds is remarkable, there are still limitations to conventional materials’ use, such as separation from the aqueous medium and the reuse of nanoadsorbents and nanocatalysts [27]. Recently, the development of nanocomposites as nanoadsorbents has been the subject of diverse research due to their increased surface area and physicochemical stability. Moreover, magnetic nanocomposites have been used as a good alternative to improve the stability, textural properties, and reuse of nanoadsorbents [28]. The facilities separate material from the aqueous medium and considerably increase their reuse, resulting in high adsorptive capacity [29]. Additionally, the same behavior is observed for magnetic nanocomposites as nanocatalysts. Using magnetic nanocatalysts allows the reuse of the material, increasing the cost-effectiveness and avoiding subsequent steps such as filtration and centrifugation [30].

When combined with other photoactive compounds, magnetic nanoparticles (MNPs) displayed prominent photocatalyst activity [31]. Moreover, the incorporation of MNPs on the surface of nanomaterials can increase surface area and affinity for the pollutant, resulting in enhanced adsorptive capacity [32]. Besides, due to their magnetic properties, the adsorbents/catalysts are easily removed from the liquid phase after reaction, without the need for centrifugation, and the use of other chemical compounds can decrease process costs [33,34,35].

Therefore, this work reports the use of nanocomposites as nanoadsorbents and nanocatalysts in the removal and degradation of pesticides, beyond the influence of diverse experimental conditions on the efficiency of these processes. Furthermore, this approach also extends to magnetic nanocomposites due to their potential and versatility in these processes. Further, the harmful effects of pesticides on the environment and public health are discussed.

The bibliographic search was performed using the Science Direct, American Chemical Society (ACS), and Royal Society of Chemistry (RSC) indexes, using Boolean logic and the following descriptors: “pesticide degradation” AND “photocatalysis” AND “nanocomposites”; “nanocomposites” AND “pesticides” AND “adsorption”. The search was limited to research article documents in the last ten years (from January 2012 to June 2022).

## 2. Contamination of Wastewater by Pesticides

Due to the considerable expansion of agriculture and extensive use of pesticides, an imbalance between water quality and quantity is observed worldwide. Furthermore, it is known that some pesticides are not completely removed from wastewater by biological treatments, mainly due to the damage these contaminants cause to the microorganisms used in the processes [36]. In addition, pesticide removal from aqueous solution by physicochemical processes, such as coagulation, flocculation, and decantation, is often inefficient [37].

Consequently, the demand for more efficient treatments has increased considerably among the processes used to remove pesticides from wastewater, especially heterogeneous photocatalysis and adsorption [38]. Advanced oxidative processes have several advantages, such as completely mineralizing pesticides and transforming them into non-toxic forms such as carbon dioxide and water [39]. Additionally, the possibility of using solar radiation as an energy source enables the photoactivation of the catalytic surface of the nanocatalyst, with a lower energy demand than the concentrations of chemical reagents applied in conventional effluent treatments [40].

Regarding adsorption, multiphasic nanoadsorbents (nanocomposites) have been widely explored due to their increased adsorptive capacity, dispersibility, and thermodynamic and physicochemical stability in aqueous media. Furthermore, nanoadsorbents are characterized as high specific surface area materials and efficient in removing organic compounds [28,41].

### Pesticides Toxicity

Table 1 shows the pesticide classification according to degree of danger, chemical structure, and toxic effects on the environment and public health.

As shown in Table 1, most pesticides are classified as moderately toxic according to the World Health Organization [42]. The listed data inform the adverse effects caused in aquatic animals, especially zebrafish. However, these effects are also observed in investigations of non-aquatic animals, microorganisms, and human cell in vitro models. Among pesticides, organophosphate compounds present greater carcinogenic, cytotoxic, and endocrine-disrupting potential. This scenario shows that the removal of these compounds from waters and wastewater is fundamental.

**Table 1 molecules-27-06261-t001:** Classification of pesticides in terms of hazardousness and chemical structure.

Pesticide	Class	Toxic Effects	Classification *	References
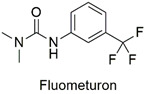	Trifluoromethyl urea	Degeneration of renal tubule epithelial cells and hemorrhage in sheep	Unlikely to have acute problems with normal use	[43]
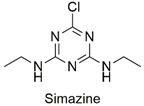	Triazine	Endocrine disruptor, carcinogen, and degeneration of renal tubule epithelial cells of *Cyprinus carpio* species	Unlikely to present a hazard in normal use	[44,45]
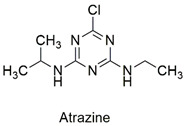	Triazine	Hematological abnormalities, degenerative and hormonal changes, cardiotoxicity, and acute microalgae toxicity	Slightly hazardous	[46,47]
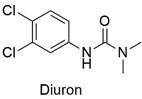	Arylurea	Acute toxicity in zebrafish embryos and abnormalities in embryonic development, reduced cell viability, and production of reactive species in the HepG2 lineage	Slightly hazardous	[48,49]
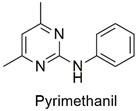	Anilinopyrimidine	Amphibian endocrine disruptor, cardiotoxicity, induction of reactive species, and apoptotic gene induction in zebrafish	Slightly hazardous	[50,51]
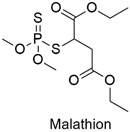	Organophosphate	Cytotoxic, carcinogenic, and genotoxic to human lymphocytes	Slightly hazardous	[52,53]
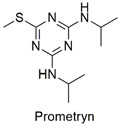	Triazine	Change in enzymatic activity and decrease in chlorophyll production in wheat crops, delay in larvae growth and carp development	Slightly hazardous	[54,55]
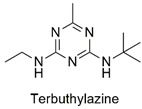	Chlorotriazine	Effect on the swimming behavior of *Danio rerio* larvae, toxic effects on human liver and kidney cells, and primary DNA damage	Slightly hazardous	[56,57]
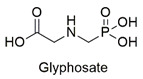	Organophosphorus	Endocrine disruptor induces human breast cancer cell growth, exposure of mice to glyphosate causes anxiety and depression-like behaviors and effects on energy metabolism of peripheral blood mononuclear cells	Slightly hazardous	[58,59,60]
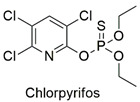	Organophosphate	Neurotoxicity, alteration in trophoblastic layer integrity, induction of ꞵ-hCG expression	Moderately hazardous	[61,62]
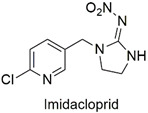	Neonicotinoid	Respiratory failure, induction of lymphocyte apoptosis, and alterations in spermatogenesis in rats	Moderately hazardous	[63,64,65]
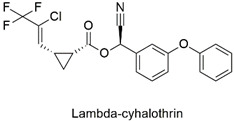	Pyrethroid	Reduced heart rate and altered thoracic limbic activity in *Daphia magna*	Moderately hazardous	[66]
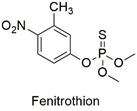	Organophosphate	Erythrocyte abnormalities in *Danio rerio* species, mutagenic and endocrine-disrupting properties	Moderately hazardous	[67,68,69]
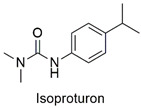	Phenylurea	Changes in the growth pattern of *Lemna minor* species reduced photosynthetic pigment production	Moderately hazardous	[70,71]
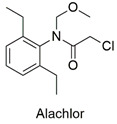	Chloroacetanilide	Production of reactive oxygen species, induction of apoptosis and damage to sperm DNA and alteration in the gene expression of the species, and decrease in cell viability of *Prorocentrum minimum*	Moderately hazardous	[72,73]
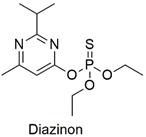	Organophosphate	Chromosomal alterations, DNA damage, carcinogenic effects, and neurotoxicity to mouse embryos	Moderately hazardous	[74,75,76]
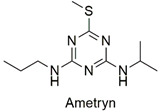	Triazine	Malformation in zebrafish larvae and oxidative stress and genotoxicity in Wistar rats	Moderately hazardous	[77,78]
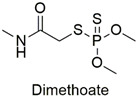	Organophosphate	Enzymatic alterations in *Cyprinus carpio* and *Galba truncatula* species	Moderately hazardous	[79,80]
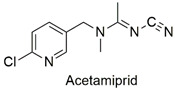	Neonicotinoid	Reduced hatchability of *Danio rerio* eggs, heart rate, and growth changes. Physiological changes in Apis *cerana cerana*	Moderately hazardous	[81,82]
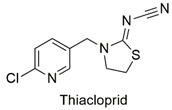	Neonicotinoid	Invertebrate toxicity, increased body mass, and liver hypertrophy in Wistar rats	Moderately hazardous	[83,84]
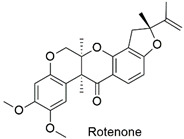	Terpene	Neurotoxic and neurodegenerative agents for humans	Moderately hazardous	[85]
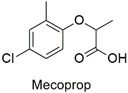	Chlorophenoxy	Toxicity to the bacterial strain of *Pseudomonas putida*	Moderately hazardous	[86]
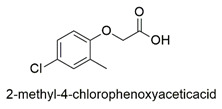	Oxi-alkanoic acid	Changes in the organization of the cell membrane of *Pseudomonas putida* and uncoupling of oxidative phosphorylation in a cell lineage of the *Metynnis roosevelti* species	Moderately hazardous	[87,88]
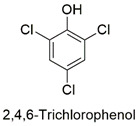	*p-*Chlorophenol	Reproductive toxicity, developmental disturbance in *Danio rerio* embryos, mutations in p53 gene from zebrafish liver	Non-cancerous	[89,90]
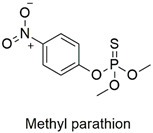	Organophosphate	Endocrine disruptor, genotoxic, cardiotoxic and neurotoxic	Highly hazardous	[91,92,93,94]
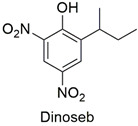	Dinitrophenol	Depletion of ATP levels, disruption of liver metabolism	Highly toxic. Obsolete as a pesticide in the European Union and the United States	[95,96]
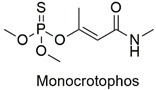	Organophosphate	Cytological anomalies in *Catla catla* fish and neurotoxicity in humans	Highly hazardous	[97,98]
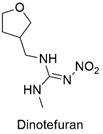	Neonicotinoid	Production of reactive species, bioaccumulation, alteration in growth and reproduction of the species *Eisenia fetida*	Acute oral toxicity	[99,100]
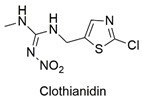	Neonicotinoid	Endocrine disruptor for the species *Eremias argus* and responsible for the loss of body mass in birds, induction of lipid peroxidation, oxidative stress, and DNA damage	Extremely toxic to aquatic life	[101,102,103]
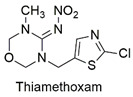	Neonicotinoid	Reduction of antioxidant activity and oxidative damage in *Chironomus ripar* and endocrine disruption in flies	Uninformed	[101,104]
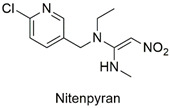	Neonicotinoid	Genotoxicity, effects on antioxidant enzymes in zebrafish and earthworm	Uninformed	[105,106]

* Adapted from Who [42].

## 3. Decontamination Methods

### 3.1. Nanocomposites

Nanocomposites are solid materials composed of two phases, the matrix and the reinforcement phase, with one of the constituents having dimensions in nanometers [22,107]. The matrix phase can be ceramic, polymeric, or metallic in origin and the reinforcement phase can be any class of nanomaterial.

From this perspective, metallic nanoparticles, graphene oxide, chitosan, and reduced graphene oxide can be used as reinforcement, which are dispersed on the matrix to generate materials with high mechanical resistance, good optical properties, and high surface area [28,108,109]. Thus, nanocomposites can be easily applied to water and wastewater treatment, catalysis, and structural applications. Furthermore, the synthesis of nanocomposites can reduce surface energy and the tendency for magnetic nanoparticles to agglomerate, increasing the physicochemical stability [110,111].

Due to cost-effectiveness and ease of application, polymeric and ceramic nanocomposites are commonly used in various applications. Polymeric-based matrix nanocomposites can be prepared by either in situ intercalation polymerization, melt intercalation of the pre-polymer solution, or sol–gel synthesis [112], while ceramic-based matrix nanocomposites can be synthesized by either sol–gel synthesis, powder process or polymer precursor process, for example [113].

#### 3.1.1. Heterogeneous Supported Nanocatalysts

Supported nanocatalysts consist of an active nanomaterial dispersed on a less active material called a catalytic support. They are widely used in heterogeneous photocatalysis due to their high surface area, considerable photocatalytic activity, and chemical/thermal stability [114].

The application of supported nanocatalysts (nanocomposites) helps overcome some drawbacks commonly encountered when applying isolated nanocatalysts, including nanoparticle agglomeration due to their high surface energy and poor dispersion in the aqueous solution [115]. Additionally, these catalytic supports increase the contact between organic pollutants and the catalytic surface [116].

Among the supported nanocatalysts used for organic pollutant degradation, TiO_2_, CuO, Fe_2_O_3_, and ZnO, supported on zeolites or silica, can be highlighted due to their significant photocatalytic activity, high surface area, and thermo-chemical stability [117]. Additionally, supported catalysts show reduced bandgap energy due to better charge separation by the presence of the support. Thus, the possibility of using this supported nanocatalyst even in the visible light range of spectrum compliments their potential applicability in light-driven processes, in particular, heterogenous photocatalysis [118]. Moreover, these nanocatalysts can be easily prepared by alternative synthetic methods, such as those based on leaves or plant extracts, microorganism strains, and industrial waste [119]. Plant-based biosynthesis of nanocatalysts is based on the reduction and stabilization of the metallic precursor. In contrast, biogenic synthesis using microorganism strains is based on a biocatalytic reduction reaction that transforms the precursor into nanoparticle suspensions, followed by nucleation and stabilization [120]. Figure 1 shows a schematic representation of the heterogeneous photocatalysis process for application in the degradation of organic pollutants.

##### Photocatalytic Degradation of Pesticides

All scientific works discussed were found in indexed platforms, and research articles were selected as the basis for investigation. Table 2 shows the main results observed in this work.

From Table 2, it was possible to observe a tendency for specific pesticide degradation, such as chlorpyrifos, atrazine, and imidacloprid. This is due to the wide use of these pesticides in agriculture, generating residues that can be released into wastewater [138,139,140].

It is worth mentioning that the utilization of magnetic nanocatalysts has been increasing once they have been characterized by improved photocatalytic activity compared to non-magnetic ones. In addition, magnetic nanocatalysts can solve operational drawbacks faced in heterogeneous photocatalysis by applying isolated nanocatalysts such as nanoparticle agglomeration and difficult recovery of the solid material after treatment [141]. Thus, in the following sections, the mechanisms involved in the pesticide degradation on support nanocatalysts will be described, as well as the enhancement of these nanocatalysts by adding a dopant material, which can be magnetic or non-magnetic.

##### Degradation of Pesticides by Heterogeneous Photocatalysis

Table 2 shows that pesticide degradation percentages are reported in the range of 80 to 90% after 30 to 180 min, with initial pesticide and nanocatalyst concentrations ranging from 15 to 50 mol L^−1^ and 0.1 to 0.5 g L^−1^, respectively. In addition, the selected scientific works report the good dispersibility of the nanocatalysts on the catalytic support, resulting in a nanocomposite with high chemical stability and preserved photocatalytic activity after 4 to 6 cycles of heterogeneous photocatalysis.

##### Effect of Initial Pesticide Concentration

Regarding the initial pesticide concentration, it is important to notice that an increase in this parameter leads to a decrease in the degradation percentage; at a high pollutant concentration, the number of pesticides molecules and active sites of the nanocomposite are altered, which results in an insufficient number of sites available to receive pesticide molecules and degrade them [142]. Therefore, an ideal initial concentration that favors photocatalytic pesticide degradation can be assumed, yielding higher oxidation percentages. In most scientific works, the concentration of 15 mol L^−1^ seems to be favorable for pesticide degradation (considering 0.1–0.3 g L^−1^) due the abundance of active sites available for pesticide molecules on the catalytic surfaces at this concentration [124,129]. In contrast, at higher concentrations (about 20 mg L^−1^ or more), pesticide molecules hindered light absorption by the nanocatalyst, which reduces hydroxyl radical generation and consequently degradation of the target pollutant [143].

##### Effect of Nanocatalyst Concentration and Dopant Incorporation

Based on the investigated scientific papers presented in Table 2, it is noticeable that an increase in the supported nanocatalyst concentration (metallic nanoparticles dispersed on catalytic support) favors the photocatalytic degradation of pesticides. This is due to the imbalance between the number of molecules of the pollutant in an aqueous solution and the number of active sites on the nanocatalyst [109]. Therefore, the constant rate values (*k*_1_, considering the Langmuir–Hinshelwood equation for the pseudo-first-order kinetic model) increase considerably, which results in greater pesticide degradation [132]. However, at extremely high nanocatalyst concentrations, pesticide degradation is reduced due to the tendency for nanocatalyst particles to agglomerate and increase the opacity of the aqueous solution, which reduces the photocatalytic activity of the supported nanocatalyst [144].

It is worth pointing out that there is a limiting value responsible for considerable pesticide degradation percentages in aqueous solution regarding dopant material. The reason for this is the same as for the increase in nanocatalyst concentration.

At the same time, it can be noticed that the incorporation of two materials (nanocomposite) can yield greater nanocatalyst surface area and greater light absorption in the electromagnetic spectrum, allowing the application of this nanocatalyst under UV and visible light radiation for heterogeneous photocatalysis.

##### Effect of pH

The solution pH has a considerable effect on pesticide degradation in heterogeneous photocatalysis. According to the scientific articles investigated (Table 2), it was possible to verify that the solution pH affects the surface charge of the nanocatalyst. The net charge of the nanocatalyst surface is measured by the point of zero charge (pH_PZC_), at which the nanocatalyst has a net charge of zero [145]. At a pH lower than pH_PZC_, the surface of the nanocatalyst is protonated and, therefore, positive. In contrast, at a pH higher than pH_PZC_, the surface of the nanocatalyst is deprotonated and, therefore, negative.

The pH interferes with pesticide degradation as it changes the charges associated with the surface of the nanocomposites used in heterogeneous photocatalysis. Thus, the surface charge is strongly related to the adsorption of pesticide molecules before their photocatalytic oxidation [29]. When the pesticide molecule has anionic character (imidacloprid), the adsorption on the surface of the nanocatalyst is favored at acidic pH. On the other hand, when the pesticide molecule has more acidic character, i.e., chlorpyrifos, the adsorption onto the catalyst surface is favored at alkaline pH, in which pH_PZC_ is greater than the solution pH.

Additionally, the solution pH can affect the reaction between hydroxyl ions and positive holes of the supported nanocatalyst and, therefore, determine a greater or lesser extent of pesticide degradation [128]. The above-mentioned reaction seems to be favored at higher solution pH (alkaline pH), resulting in more hydroxyl radicals, which are responsible for pesticide photocatalytic degradation.

##### Effect of Temperature

In some scientific research involving heterogeneous photocatalysis and pesticide degradation (Table 2), the effect of temperature on pesticide photocatalytic degradation was investigated. The effects of temperature on heterogeneous photocatalysis using supported nanocatalysts, magnetic or non-magnetic, are associated with the energy required for the adsorption and desorption of pesticide molecules. Some studies found that the adsorption of imidacloprid was facilitated by increasing the temperature from 20 to 80 ° C. Due to the endothermic nature of the pesticide adsorption–degradation process, which is the majority of cases studied, pesticide absorption on the catalytic surface and adsorption rate increase with temperature [131]. In addition, the degradation rate seems to be increased at temperatures above the atmospheric temperature (i.e., 25 ± 5 °C).

##### Effect of Nanocatalyst Reuse

In most scientific research involving heterogeneous photocatalysis and pesticide degradation (Table 2), the effect of nanocomposite reuse is evaluated. According to the results presented in the selected studies, the combination of two or more nanomaterials in the nanocatalyst composition yields greater photocatalytic activity, even after five cycles of reuse. However, it is worth pointing out that when one of the nanocomposite phases has magnetic properties, its efficiency in pesticide degradation improves considerably after various photocatalytic cycles. Therefore, magnetic nanomaterials have been highlighted due to their versatility, ease of operation (magnetic separation), and reuse capacity. In this view, about 72 to 80% of pesticide degradation is reported even after six to seven cycles of heterogeneous photocatalysis [128,129].

#### 3.1.2. Adsorption

Adsorption is a physicochemical process and surface phenomenon wherein a fluid (liquid or gas) interacts with a solid surface (adsorbent), resulting in the mass transfer of a solute from a fluid phase to the solid surface [146]. In this system, the degree of interaction of ions and molecules depends on the concentration, pH, temperature, and available specific surface are [147]. The attractive forces between the adsorbent and the adsorbate can be divided into physical and chemical adsorption [148,149].

Chemical adsorption (chemisorption) involves the transfer of electrons and chemical bond formation. On the other hand, physical adsorption results from weak intermolecular interactions called van der Waals forces, electrostatic interactions, H-bonds, and π–π bonds [150].

Regarding the nature of adsorption, some parameters must be observed, such as adsorbent selectivity, homogeneity, and heterogeneity of the solid nanomaterial and the adsorption rate, which can be fast or slow. Furthermore, depending on the process, the adsorbate can accumulate at the adsorbent interface in a monolayer or multilayer [151]. For a further explanation, Figure 2 illustrates the adsorption mechanisms in mono and multilayer models.

Adsorption is a relatively simple technology with low cost and energy demand and can effectively remove a range of chemical compounds from aqueous medium. However, the effectiveness of the process depends on the intrinsic properties of adsorbents, such as particle diameter, pore volume, and chemical/thermal stability [152]. In this context, diverse strategies have been explored, such as the use of thermal treatments, modification, and surface doping that aim not only to increase the adsorptive capacity but also to reduce costs associated with adsorbent synthesis [153,154]. For instance, 70% of the cost of an adsorption operation is relative to the adsorbent [155].

##### Adsorption of Pesticides into Nanocomposites

The research was performed (Table 3) to verify the efficiency of nanocomposites used as nanoadsorbents.

Based on the results presented in Table 3, it was possible to verify that the pesticide adsorption studies using nanocomposites were efficient for the removal of organic compounds from aqueous solution, especially in the class of organophosphates neonicotinoids and triazines.

Moreover, the studies were heterogeneous enough and contemplated different compounds belonging to the same chemical class. However, in most articles, the nanoadsorbent showed selectivity and higher adsorption capacity when compared to isolated materials, i.e., applied individually. Hybrid materials were usually synthesized to combine individual properties and make them multifunctional [112].

In this study, it was found that nanocomposites have good performance in the adsorption process. In addition, the efficiency and stability of the adsorbent were maintained after several adsorption/desorption cycles. Furthermore, because some materials exhibit magnetic behavior, the adsorbent can be easily separated from the solution by the application of an external magnetic field. Figure 3 shows a simple approach to magnetic nanocomposites used for pesticide adsorption, magnetic retention, removal, and reuse.

##### Effect of Initial Adsorbate Concentration and Adsorbent Dosage

Studying the effect of experimental parameters such as initial adsorbate concentration, adsorbent dosage, and adsorption contact time is essential to determine the most favorable conditions for contaminant removal and to investigate the phenomenon of adsorption.

Muda et al. [162] demonstrated that the maximum adsorption capacity was directly proportional to the increase in the initial concentration of adsorbate and adsorbent mass, mainly due to the high probability of adsorbate molecules binding to the adsorption sites [160]. Likewise, the gradual increase in the initial concentration of adsorbate and the amount of adsorbent can reduce the contaminant removal percentage due to the saturation of higher energy sites and the interaction between the adsorbent molecules [165].

Another important factor for the adsorption efficiency is the contact time between adsorbent and adsorbate. In most of the studies, adsorption showed distinct behavior over time. In the initial stages, a fast adsorption rate is observed (mass transfer of liquid phase to the solid phase) and a tendency to plateau in the subsequent steps because of saturation of the adsorption sites (equilibrium between vacancy sites and adsorbed molecules) [145]. At the same time, the nanocomposite content (ratio mass: mass) of the of constituent materials can, in some cases, affect the efficiency of the process, either positively or negatively. For example, Gupta et al. [166] reported that the combination of cellulose nanofibers and cadmium sulfide resulted in more efficient nanomaterials for removing chlorpyrifos, mainly due to the increase in the specific area and, consequently, the greater number of adsorption sites available. However, the increase in biopolymer charge (>10%) significantly reduced the adsorbent performance due to the aggregation of particles and the steric hindrance promoted by the functional groups present in the polymeric chains.

##### Effect of pH

The pH of the solution is an important parameter that influences the adsorption capacity since it is related to the surface properties of the adsorbent, degree of speciation of chemical compounds, and competition of ions in solution for adsorbent sites [28,161]. Nevertheless, the ideal pH for each process is quite specific, as it depends on the adsorbent’s surface charge and the adsorbate’s intrinsic characteristics, such as the acid dissociation constant (pK_a_) and chemical stability to pH changes. In addition, the study of pH effects can also help to understand the possible chemical or physical mechanisms involved in the removal of the adsorbate [167,168].

Recently, Abukhadra et al. [160] developed a polymeric nanocomposite containing bentonite for methyl-parathion removal. The study of the effect of pH verified a considerable increase in the adsorption capacity in alkaline medium (pH = 9.0). This adsorption behavior is related to the point of zero charge (pH_PZC_) of the adsorbent and deprotonation at basic pH, which results in strong electrostatic attraction with the cationic species of methyl-parathion.

However, van der Waals forces are not the only interactions existing between the adsorbent and the adsorbate since the pH of the solution may not influence the process [145]. According to the chemical structure of compounds and functional groups, the interaction between adsorbent/adsorbate systems can occur through stacking bonds π–π, hydrogen bonds, and ionic exchange [156,161].

##### Effect of Temperature on Adsorption

The efficiency of the adsorption process is highly dependent on experimental conditions, such as temperature. Evaluating the effect of temperature on adsorption behavior is an important step considering a future application in wastewater treatment systems [111].

In an adsorption system, the temperature can affect the adsorption velocity and kinetic energy, resulting in an increase or decrease in the removal percentage and adsorption capacity [156,169]. The effects of temperature on adsorption occur due to viscosity, rate diffusion, and equilibrium state [170].

Adsorption is a spontaneous process that can be endothermic or exothermic [171,172]. Observations concerning the temperature dependence of adsorption of pesticides were reported. All studies that evaluated the effect of temperature on adsorption assumed an endothermic reaction, i.e., the reaction became more favorable at higher temperatures [156,157,159,160].

##### Equilibrium and Adsorption Kinetics

Adsorption isotherms are helpful mathematical equations to describe the equilibrium of adsorption; that is, the amount of solute adsorbed on the surface of the adsorbent as a function of concentration in the liquid phase at a constant temperature [173]. Furthermore, they are relevant tools for understanding the interaction between the adsorbent and the adsorbate, which can help process design [174]. Several isothermal models can describe the adsorption phenomenon. However, the most commonly reported in the literature are the Langmuir, Freundlich, and Temkin isotherms.

The Langmuir isotherm is a theoretical model used to describe processes on a uniform, non-porous surface. The model is based on the hypothesis that adsorption occurs in monolayers with uniform energy distribution, in which the molecules of the adsorbed solutes do not interact with each other [175]. The expression of the Langmuir isotherm is represented by Equation (1):(1)$$q_{e}=\frac{q_{\max}\;C_{e}}{{1+K}_{L}\;C_{e}}$$
where *q_e_* is the amount of adsorbate adsorbed per unit of mass of adsorbent in equilibrium (mg g^−1^), *q_m_* is the maximum adsorbed amount (mg g^−^¹), *K_L_* is the adsorption equilibrium constant or Langmuir constant (L mg^−1^), and *C_e_* is the concentration of adsorbate in the solution at equilibrium (mg L^−1^).

The Freundlich isotherm is an empirical mathematical model that represents the adsorption equilibrium of a fluid (liquid or gas) on the surface of a solid material. This equation assumes that the adsorption occurs on heterogeneous surfaces and adsorption sites of different energy levels [176]. The Freundlich isotherm is expressed by Equation (2):(2)$$q_{e}{=K}_{F}{\;(C}_{e}{)\;}^{\frac{1}{n}}\;\;\;$$
where *K_F_* is the adsorption equilibrium constant or Freundlich constant ((mg g^−1^) (L mg^−1^)^−1/*n*^), and 1/*n* is the heterogeneity factor.

The Temkin isotherm is an empirical model used to describe the indirect interactions between the adsorbent and the adsorbate and assumes that the heat of adsorption decreases as the surface of the adsorbent is covered. Unlike the Freundlich isothermal model, the system energy decreases in a non-logarithmic form (uniform energy distribution) [177]. Temkin’s model is given by Equation (3) and is applied in studies to measure the energetic contributions of the adsorption process. Thus, the process temperature must be considered.
(3)$$q_{e}=\frac{\mathrm{RT}}{b}\ln\left(K_{T}C_{e}\right)$$
where *B* is the constant associated with adsorption heat (kJ mol^−1^), *T* is the temperature (K), *R* is the universal constant of ideal gases (8.314 J mol^−1^ K^−1^), *b* is the constant associated with the interaction between adsorbate and adsorbent, and *K_T_* is the Temkin constant (L mg^−1^).

Furthermore, the Dubinin–Radushkevich (DR) model is often reported as a good fit for the experimental data to the adsorption isotherms [178]. However, this empirical model assumes active sites with Gaussian energy distribution and is widely applied only to intermediate ranges of adsorbate concentration, as it presents an unrealistic behavior [179].

Kinetic models have been widely used to assess adsorbent performance, construct the kinetic profile, and investigate the possible mechanisms involved in mass transfer between liquid and solid phases [180]. Among the proposed models, the most common are pseudo-first-order (PFO), pseudo-second-order (PSO), and Elovich kinetics [181,182].

The PFO model was initially proposed by Lagergren and describes that the adsorption process is independent of the initial concentration of adsorbate and which adsorbent has some active sites, with the external/internal diffusion being the step determining the adsorption rate. Moreover, the model suggests that the interactions between the molecules of the adsorbent and the adsorbed substance are relatively weak attractive forces, indicating a physisorption process. The PFO kinetics is based on the non-competition of molecules for an active site (1:1 ratio) and is expressed by Equation (4):(4)$$q_{t}{=q}_{1}\left({1-e}^{{-k}_{1}\cdot t}\right)$$
where *q_t_* is the amount of solute adsorbed on the solid surface at time *t* (mg g^−1^), and *k*_1_ is the pseudo-first-order kinetic constant (min^−1^).

The PSO model assumes that adsorption on active sites is the control step of the adsorption process, where the adsorption rate increases proportionally with the available adsorption sites. Thus, adsorption occurs in specific sites, and adsorbate molecules do not interact with each other. In terms of adsorption mechanisms, if this model presents the best adjustment of the experimental data (higher values of *R^2^*), the electrons transfer is the limiting rate of the process, indicating a phenomenon of a chemical nature. The pseudo-second-order kinetics, however, consider the competition between the adsorbate molecules for an active site (2:1 ratio) and are expressed by Equation (5):(5)$$q_{t}=\frac{t}{\left(\frac{1}{k_{2}q_{2}^{2}}\right)+\left(\frac{t}{q_{2}}\right)}$$
where *k*_2_ is the pseudo-second-order kinetic constant (g mg^−1^ min^−1^), and *t* is the contact time (min)

Elovich model is largely used to describe the chemisorption of solutes onto solid sorbents and can be expressed by Equation (6):(6)$$q_{t}=\frac{1}{b}\ln\left(\mathrm{abt}\right)$$
where *a* is the constant that describes the initial sorption rate (mg g^−1^ min^−1^) and *b* is the constant related to surface coverage and activated energy for chemisorption (g mg^−1^).

From the scientific articles shown in Table 3, it was possible to verify that the kinetic models of PPO and PSO are the most used to measure the adsorption rates of pesticides on nanoadsorbents (magnetic and non-magnetic). Regarding the adsorption equilibrium models, the Langmuir and Freundlich isotherms are applied in adsorption studies in the aqueous phase to represent a good adjustment of the experimental data.

The Langmuir isotherm was reported as the better model for adjusting the equilibrium of the pesticide’s adsorption, suggesting adsorption in monolayers with uniform distribution of active sites of nanocomposites. Furthermore, it is assumed that there is no competition between the adsorbent molecules for a single active site of the adsorbent.

Concerning the kinetic models, the PSO model demonstrated the best adjustment of experimental data, indicating that mass transfer as a function of a concentration gradient is the limiting step of adsorption. Furthermore, the number adsorbate and adsorbent sites had a 2:1 relationship [183]. Furthermore, this model suggests that the transfer of pesticides to the solid surface occurs predominantly by chemical mechanisms, i.e., chemisorption.

## 4. Conclusions and Future Perspectives

The present work made it possible to verify the potential and extensive application of magnetic and conventional nanomaterials in either degradation or removal of pesticides from the aqueous solution. Thus, it is observed that the degradation of these pollutants occurs mainly by heterogeneous photocatalysis, whereas the removal of wastewater is mainly by adsorption processes.

The pesticides methyl-parathion, dinoseb, and monocrotophos are the most dangerous. However, photodegradation and adsorption are effective for moderately toxic pollutants, such as chlorpyrifos and atrazine. Values of up to 90 and 99% are reported for degradation and the removal of pesticides from wastewater. This result demonstrates a certain lack of research involving the adsorption or photodegradation of highly hazardous pesticides.

At the same time, the wide-ranging use of magnetic nanocomposites is justified by their operational ease, reuse, and chemical or thermal stability. In addition, the percentage of materials used in the synthesis of nanoadsorbents can affect the interaction between the adsorbent and the adsorbate. The pH of the solution, initial concentration of adsorbate, contact time, and surface area are factors determining the efficiency and optimization of the process.

Photocatalytic degradation is reported under visible light and UV radiation due to the addition of different nanomaterials that make up the reinforcement layer, and the nanocomposite matrix is combined. Further, the catalytic efficiency is considerably enhanced by the incorporation two materials in the nanocatalyst structure. Therefore, it was possible to show the potential of nanocomposites in the degradation/removal of pesticides in aqueous solutions.

Studies involving the scale-up of adsorption and photocatalytic processes should be encouraged, considering the higher reuse capabilities and efficiency of nanocomposites, especially the magnetic ones. In this view, heterogeneous photocatalysis for the degradation of pesticides at higher concentrations and adsorption of these pollutants in fixed-bed adsorbers should be conducted to expand the breakthrough performed by the research discussed in this review. In addition, the search for alternative sources to produce nanocomposite matrixes, such as residual or abundant materials (biomass), can intensify the use of these nanocomposites in heterogeneous photocatalysis and the adsorption of pesticides. At the same time, ecotoxicity and cytotoxicity studies should be performed to assess the safety of the use of these materials and their implementation at a large scale.

## Figures and Tables

**Figure 1 molecules-27-06261-f001:**
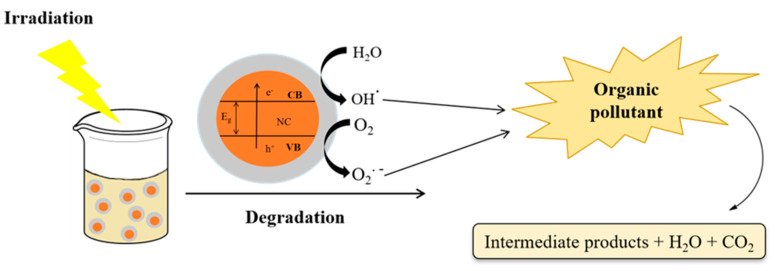
Scheme showing degradation of organic pollutants by heterogeneous photocatalysis.

**Figure 2 molecules-27-06261-f002:**
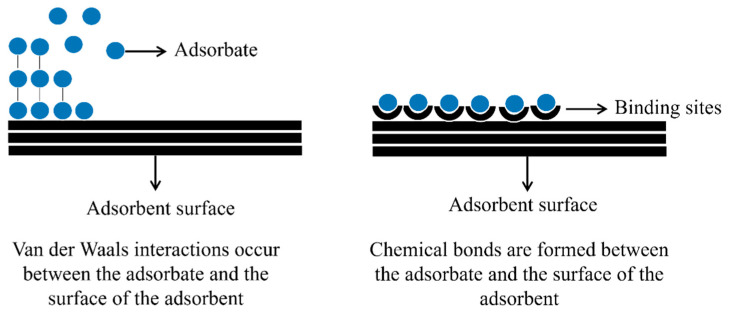
Illustration of the adsorption process in mono and multilayer.

**Figure 3 molecules-27-06261-f003:**
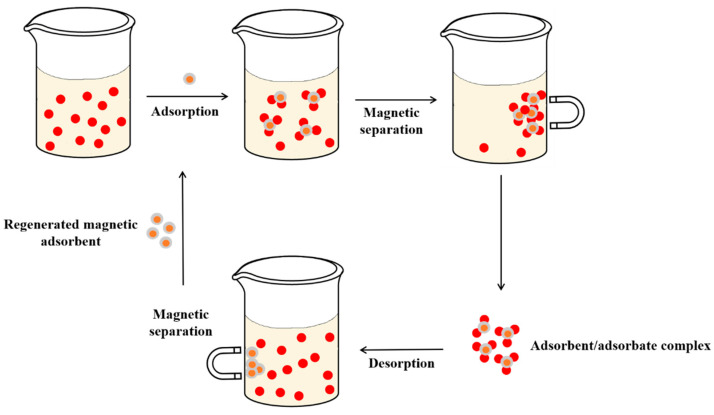
Use of magnetic nanoadsorbents in liquid phase adsorption.

**Table 2 molecules-27-06261-t002:** Photocatalytic degradation of pesticides using supported nanocatalysts.

Research	Pesticide	Operational Conditions	Comments	Reference
Photocatalytic degradation of pesticides under visible light using TiO_2_@rGO nanocomposite	Atrazine, isoproturon, alachlor, diuron	50 mol L^−1^ of pollutant, 0.25 g L^−1^ catalyst, pH ~6. Pesticide degradation greater than 80% after 180 min.	rGO incorporation resulted in anatase formation to a higher extent than the rutile phase.	[121]
Catalytic photodegradation of imidacloprid on C_3_N_4_ doped with H_2_O_2_, fullerene, and functionalized with P	Imidacloprid	0.6 g L^−1^ of catalyst. 91–95% of degradation after 12 h.	Good chemical stability and recyclability of the nanocatalyst. H_2_O_2_ addition favored HO• formation.	[122]
GO nanosheets decorated with CuFe_2_O_4_ and CdS nanoparticles as heterogeneous photocatalyst	Imidacloprid and Dinoseb	15 mol L^−1^ of pollutant, 0,15 g L^−1^ of catalyst, pH ~7. Degradation percentage ranged from 91–94% after 140 min.	Utilization in magnetic separation for nanocatalyst collection. Good performance in pesticide degradation.	[123]
Synthesis and characterization of ZnO@CoFe_2_O_4_ magnetic nanocomposite for pesticide photocatalytic degradation.	Imidacloprid	15 mol L^−1^ of pollutant, 0.1 g L^−1^ of catalyst, pH 10. 79.3% degradation after 45 min.	Good nanoparticle dispersion on the catalytic supported. Photocatalytic activity enhancement.	[124]
Synthesis of TiO_2_@chitosan and MOF (MIL-88(Fe)) for pesticide degradation	Organophosphate pesticides	98.79% degradation under visible light after 30 min.	Excellent photostability of the nanocatalyst after 5 cycles of reuse.	[125]
Fe_3_O_4_@TiO_2_-Graphene magnetic nanocomposite for colorimetric detection and pesticide photocatalytic degradation in aqueous media.	Atrazine	15 mol L^−1^ of pollutant, 0.50 g L^−1^ of catalyst, pH 7. Pesticide degradation of 90% after 120 min.	Good dispersion of TiO_2_ on graphene. Excellent nanocatalyst photocatalytic activity.	[126]
Photoluminescence emission behavior in the bandgap reduction of CeO_2_@SiO_2_, with Fe doping.	Chlorpyriphos	50 mol L^−1^ of pollutant, 0.5 g L^−1^ of nanocatalyst, pH ~10. 81.31% degradation after 180 min.	Drastic bandgap energy of the nanocomposite (3.77 to 2.22 eV).	[127]
Ni-Co nanocatalyst doped with S decorated with Fe_3_O_4_ nanoparticles.	Chlorpyriphos	2.5 mg L^−1^ of pollutant, 60 mg L^−1^ of catalyst, pH ~10. Pesticide degradation of 92.5% after 150 min.	Magnetic nanoparticle incorporation resulted in higher catalytic efficiency, even after 7 cycles of heterogeneous photocatalysis.	[128]
Fe_3_O_4_@CdS-ZnS magnetic nanocomposite for dyes and photocatalytic degradation of pesticides.	Chlorpyrifos	15 mol L^−1^ of pollutant, 0.01 g L^−1^ of catalyst, pH 10. Pesticide degradation of 95% after 180 min.	72% degradation of the pesticide after 6 cycles of heterogenous photocatalysis.	[129]
Green synthesis of ZnO @ CdS nanocomposite for pesticide photocatalytic degradation.	Atrazine and chlorpyrifos	Pesticide degradation in the range of 89–91% after 150 min.	High surface area (111 m^2^ g^−1^).	[130]
Pesticide photocatalytic degradation on Fe_3_ O_4_@GO-TiO_2_ and NiO under visible light.	Imidacloprid	15 mol L^−1^ of pollutant, 0.1 g L^−1^ of catalyst, pH 7. Degradation of 81% after 45 min.	Reduction of 6% of the photocatalytic activity after 4 cycles of heterogenous photocatalysis.	[131]
ZnO@Bi_2_O_3_ nanocomposite modified by surfactant for degradation pesticide under visible light. l	Lambda-cyhalothrin	30 mol L^−1^ of pollutant, 1.2 mg L^−1^ of catalyst at pH 7.79% of degradation after 120 min.	Excellent photocatalytic activity of the nanocomposite under visible light.	[132]
Morphologic influence of CuNPs@ZnO nanocomposite on photocatalytic degradation.	Methyl-parathion	99% of pesticide degradation after 80 min.	Excellent selectivity for methyl-parathion.	[133]
Carbon nitride functionalized with triethanolamine for pesticide photocatalytic degradation.	Atrazine	90% of atrazine degradation under UV radiation after 60 min.	Simultaneous removal of 10 pesticides, including bactericides, herbicides, and fungicides.	[134]
TiO_2_-Ag_3_PO_4_ nanocomposite application in pesticides photocatalytic degradation.	Atrazine, Imidacloprid, and Pyrimethanil	10 mg L^−1^ of pollutant, 0.5 g L^−1^ of catalyst. Degradation percentages were in the range of 25–87%.	The nanocomposite showed higher photocatalytic activity than the isolated compounds.	[135]
The ternary semiconductor is used for organophosphate degradation.	Malathion, monocrotophos, and chlorpyriphos	3 mg L^−1^ of photonanocatalyst. Pesticide photocatalytic degradation ranged from 94 to 97% after 60 min.	The nanocatalyst showed excellent photocatalytic activity after various heterogeneous photocatalytic cycles.	[136]
Ag-ZnO nanocomposite for chlorpyrifos degradation.	Chlorpyrifos	500 mol L^−1^ of pesticide, 0.2 g L^−1^ of catalyst. Pesticide degradation ranged from 90, 75, and 65% for Ag-ZnO, using 3% Ag-ZnO and 1% ZnO.	The nanocatalyst containing 3% silver resulted in greater photocatalytic activity.	[137]

**Table 3 molecules-27-06261-t003:** Pesticide adsorption by magnetic and non-magnetic nanocomposites.

Study	Pesticide	Experimental Conditions	Comments	Reference
Synthesis of graphene oxide hybrid microspheres and polyvinylpyrrolidone cross for organochlorine pesticide adsorption	2,4,6-Trichlorophenol	The initial concentration of adsorbate (50–300 mg L^−1^), amount of adsorbent (0.5–4 g L^−1^), solution pH (2–10), and thermodynamic study (15, 25, 30 °C).	The nanocomposite has an excellent adsorptive capacity (466.7 mg g^−1^). The adsorption occurs by chemical mechanisms, in which the π–π interactions and hydrogen interactions are contemplated.	[156]
Magnetic nanocomposite with a metal–organic structure for the removal of organophosphate pesticides	Fenitrothion	Initial concentration of adsorbate (5–−50 mg L^−1^), solution pH (3–10), adsorbent concentration (5–40 mg L^−1^). An ideal condition for the pesticide removal was pH 7.0, using 30 mg L^−1^ of adsorbent, and 10 mg L^−1^ of the adsorbate.	The type better described the fenitrothion adsorption study Langmuir I isotherm and pseudo-second-order kinetics, indicating chemical adsorption.	[157]
Synthesis of hybrid magnetic nanocomposite for the removal of organophosphate pesticides	Phosphamidon, chlorpyrifos, diazinon, dimethoate	Adsorbent dosage (10–120 mg), initial concentration of pesticides (10–100 µg mL^−1^), and study of the influence of pH (3–11).	The nanocomposite exhibits greater pesticide adsorption capacity when compared to non-hybrid compounds. The q_max_ ranged between 37.18–76.34 mg g^−1^, and the adsorbent showed a high affinity for the pesticide diazinon.	[158]
Magnetic reduced graphene oxide nanocomposite for triazine removal	Ametrine, promethrin, simazine and atrazine	Study the influence of pH, temperature, adsorbent dosage, and contact time.	The magnetic nanocomposite is effective in removing pesticides. Maximum capacity was reached at pH 5.0. The ions present in the solution, increase in temperature, and initial adsorbate concentration caused a significant increase in the adsorptive capacity. Due to its magnetic behavior, the absorbent can be easily recovered and proved effective after 7 cycles of the process.	[159]
Hybrid nanocomposite intercalated with bentonite as adsorbent for the pesticide methyl-parathion	Methyl-parathion	pH effect (2–9), initial concentration of adsorbate (200–1400 mg L^−1^), adsorbent mass (0.4–1.6 g L^1^), and thermodynamic study (25–50 °C).	The adsorption capacity is pH-dependent and increases proportionally with the pH of the system, contact time with the adsorbent, initial adsorbate concentration, and temperature. The pollutant adsorption is an endothermic process, spontaneous, with experimental data represented by the Freundlich isotherm and pseudo-second-order model. The nanoadsorbent proved to be efficient in removing about 97% of the pesticide after five adsorption cycles.	[160]
Magnetic graphene oxide/ꞵ-cyclodextrin nanocomposite for neonicotinoid pesticide adsorption	Thiamethoxam, imidacloprid, acetamiprid, nitenpyram, dinotefuran, clothianidin, and thiacloprid	The initial concentration of insecticides (0.5–100 mg L^−1^) and adsorbent dosage (5.0 g L^−1^).	The adsorbent showed a greater affinity for the insecticide imidacloprid, but the q_max_ values are similar. The mechanisms involved in adsorption include hydrophobic interactions, hydrogen bonds, and the stacking of π–π bonds between the adsorbent/adsorbate complex.	[161]
Application of chitosan-graphene oxide nanocomposite as rotenone adsorbent	Rotenone	The initial concentration of rotenone (10–100 mg L^−1^), solution pH (1–9), and adsorbent containing different mass proportions of graphene oxide (CS-GO 1%, CS-GO 2%, CS-GO 3%).	The adsorption capacity decreased at alkaline pH. However, a considerable increase in the q_max_ was observed with nanocomposites containing a greater amount of GO, corroborating the hypothesis with more adsorption sites available.	[162]
Synthesis of hexadimethrine-montmorillonite nanocomposite for application as a pesticide adsorbent	Fluometuron, diuron, terbuthylazine, simazine, mecoprop, and 2-methyl chlorophenoxyaceticacid	The initial concentration of adsorbate (1.0 mg L^−1^), 20 mg L^−1^ of adsorbent, and contact time (24 h).	The adsorbent demonstrated a greater affinity for anionic compounds. Removal percentages ranged from 54 to 75%.	[163]
Montmorillonite/structured carbon nanocomposite for pesticide removal	Chlorpyrifos	The initial concentration of adsorbate and adsorbent dosage (8.0 g L^−1^), contact time (24 h), no pH adjustment	Pristine montmorillonite showed greater pesticide adsorption capacity. The possible mechanism involved in the adsorption of the organic compound is the physical interaction with the surface of the adsorbent.	[164]
Removal of permethrin from aqueous solutions using chitosan/zinc oxide nanocomposite	Permethrin	Initial concentration of adsorbate (0.05–2.5 mg L^−1^), adsorbent amount (0.01–1.5 g L^−1^), contact time (45 min), solution pH (3–11), and temperature at 25 °C	The CS-ZnO nanocomposite was able to remove about 96% of the pesticide. Adsorbent concentration, initial concentration of adsorbate, contact time, and solution pH influence the removal percentage.	[165]
Cadmium Sulfide Doped Cellulose Nanofibers for Pesticide Removal	Chlorpyrifos	Influence of contact time, pH (3–11), adsorbent dosage (0.5–2 g L^−1^), percentage of the polymer matrix in the nanocomposite (5, 10,15 20, 50%), initial concentration of adsorbate (1–8 mol L^−1^)	The removal percentage increases proportionally with the amount of adsorbent and adsorbate. However, it decreases with increasing pH, mainly due to electrostatic repulsion. Adsorption of chlorpyrifos occurs in monolayers and with uniform energy distribution.	[166]
Synthesis of SiO_2_@Fe_3_O_4_@GO-phenylethylamine nanocomposite for removal of organophosphate pesticides	Chlorpyrifos, malathion, and parathion	pH effect (3–11), adsorbent dosage (2–40 mg L^−1^), initial concentration of adsorbate (3–50 mg L^−1^)	The nanoadsorbent showed good performance after 10 adsorption cycles.	[145]

## Data Availability

Nor applicable.

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
