# Peer review of "Adsorption and Photocatalytic Degradation of Pesticides into Nanocomposites: A Review"

_molecules, 2022, doi:10.3390/molecules27196261_

Round 1

Reviewer 1 Report

The problem of decontamination of waters polluted with persistent organic pollutants is a great challenge for modern science. The application of innovative methods for this purpose, including those involving different nanomaterials, is increasingly actively promoted by specialists in the field. However, the critical mass of knowledge in this area, necessary for a qualitative leap, has not yet accumulated.

From this point of view, a review, which concentrates the achievements obtained recently by researchers in the field of adsorption and photocatalytic degradation of pesticides into nanocomposites and supported heterogeneous nanocatalysts is welcome.

The review analyzes the latest results obtained by researchers from different scientific centers and helps enormously to appreciate the current level reached in this field.

Highly appreciating the authors' effort, I have some suggestions for improving the analyzed review:

1. Authors should present a description of the applied research methodology: at least the purpose of the analysis of the publications and the principles for selecting the publications included in the analysis. This information should be included in both the abstract and the introduction.

2. In Table 1, which is entitled Classification of pesticides in terms of hazardousness and chemical structure, it would be good if the pesticides were arranged in order of increasing or decreasing hazardousness, this being the main indicator of the danger these substances present.

3. Subchapters 2.2 and 2.3 and 2.6 (there is a numbering error here) should be removed from the composition of chapter 2, because they do not refer to Water Contamination with pesticides but to decontamination methods. These subchapters would do well to compose another chapter with a suggestive title.

4. The title of the Results and discussions section is suitable for an article that describes an experimental research of its own, in the case of a review, the title should be changed.

5. In table 2 and 3 I recommend separating the researches analyzed by horizontal lines to facilitate reading.

6. I recommend completing the article with the limitations of the analyzed methods and prospective research directions in this field.

Author Response

Manuscript ID molecules-1908464

Previous Title: Adsorption and photocatalytic degradation of pesticides into nanocomposites: a review

Corresponding author: Luis F. O. Silva, William L. Silva, Guilherme L. Dotto, and Cristiano R. B. Rhoden

Listed co-author(s): Franciele S. Bruckmann, Carlos Schnorr, Leandro R. Oviedo, Luis F. O. Silva, William L. Silva, Guilherme L. Dotto, Cristiano R. Bohn Rhoden

Ms. Jolin Xie X.J.

MDPI Office

We thank you for the opportunity to submit our revised paper to the Molecules. We revised the manuscript with great attention. The Referee's comments provide us with detailed and very useful reports. The modifications are highlighted in yellow in the final version of the manuscript. By addressing their comments in the revised version, we are confident that the paper has improved considerably, thus shaping it into a publishable form. Please, find below the answers to the Referee's comments.

With kind regards

Prof. Dr. Guilherme L. Dotto

Referee #1

The problem of decontamination of waters polluted with persistent organic pollutants is a great challenge for modern science. The application of innovative methods for this purpose, including those involving different nanomaterials, is increasingly actively promoted by specialists in the field.

However, the critical mass of knowledge in this area, necessary for a qualitative leap, has not yet accumulated. From this point of view, a review, which concentrates the achievements obtained recently by researchers in the field of adsorption and photocatalytic degradation of pesticides into nanocomposites and supported heterogeneous nanocatalysts is welcome.

The review analyzes the latest results obtained by researchers from different scientific centers and helps enormously to appreciate the current level reached in this field.

Highly appreciating the authors; effort, I have some suggestions for improving the analyzed review.

Authors: We thank the Referee for our paper evaluation, and we consider the comments and suggestions to improve the paper a lot.

Queries are given in black and answers in blue.

  1. Authors should present a description of the applied research methodology: at least the purpose of the analysis of the publications and the principles for selecting the publications included in the analysis. This information should be included in both the abstract and the introduction.

A: Thanks for the observation. The applied research methodology was added in both the abstract and the introduction. See main manuscript (lines 36-41 and 136-142).

  1. In Table 1, which is entitled Classification of pesticides in terms of hazardousness and chemical structure, it would be good if the pesticides were arranged in order of increasing or decreasing hazardousness, this being the main indicator of the danger these substances present.

A: Thank you for the suggestion. The correction was made (see the main manuscript, line 176).

  1. Subchapters 2.2 and 2.3 and 2.6 (there is a numbering error here) should be removed from the composition of chapter 2, because they do not refer to Water Contamination with pesticides but to decontamination methods. These subchapters would do well to compose another chapter with a suggestive title.

A: Sections were organized, and subchapters were restructured. Thank you for the observation.

  1. The title of the Results and discussions section is suitable for an article that describes an experimental research of its own, in the case of a review, the title should be changed.

A: Thank you, it was removed.

  1. In table 2 and 3 I recommend separating the researches analyzed by horizontal lines to facilitate reading.

A: Horizontal lines were added to Tables 2 and 3 (See main manuscript, lines 234 and 369).

  1. I recommend completing the article with the limitations of the analyzed methods and prospective research directions in this field.

A: Considering the number of publications coming up, we understand that the greatest limitation should be the focus of researchers around the world should be the scale-up from the recent diverse methodologies reported. Besides, studies on removing multiple contaminants using common adsorbents/photocatalysts.

Reviewer 2 Report

This article summarizes the application of conventional nanocomposites and magnetic nanocomposites for pesticide adsorption and photocatalytic degradation in aqueous solutions. The authors are very detailed in their discussion about the types of materials, working principles and research conditions, which can make it easier for the reader to understand the status of the field. This article can be accepted if the following questions can be answered.

1. What are the advantages and disadvantages of current materials other than nanocomposites? In comparison, what are the advantages of nanocomposites?

2. In this study, why are conventional nanocomposites and magnetic nanocomposites divided into two categories? Is it possible to divide them according to other criteria, such as photocatalytic nanomaterials?

3. In the conclusion section, the authors should indicate the future directions.

Author Response

Manuscript ID molecules-1908464

Previous Title: Adsorption and photocatalytic degradation of pesticides into nanocomposites: a review

Corresponding author: Luis F. O. Silva, William L. Silva, Guilherme L. Dotto, and Cristiano R. B. Rhoden

Listed co-author(s): Franciele S. Bruckmann, Carlos Schnorr, Leandro R. Oviedo, Luis F. O. Silva, William L. Silva, Guilherme L. Dotto, Cristiano R. Bohn Rhoden

Ms. Jolin Xie X.J.

MDPI Office

We thank you for the opportunity to submit our revised paper to the Molecules. We revised the manuscript with great attention. The Referee's comments provide us with detailed and very useful reports. The modifications are highlighted in yellow in the final version of the manuscript. By addressing their comments in the revised version, we are confident that the paper has improved considerably, thus shaping it into a publishable form. Please, find below the answers to the Referee's comments.

With kind regards

Prof. Dr. Guilherme L. Dotto

Referee #2

This article summarizes the application of conventional nanocomposites and magnetic nanocomposites for pesticide adsorption and photocatalytic degradation in aqueous solutions. The authors are very detailed in their discussion about the types of materials, working principles and research conditions, which can make it easier for the reader to understand the status of the field. This article can be accepted if the following questions can be answered.

Comments to the Authors:

Authors: We thank the Referee for the positive evaluation of our paper, and we consider all the comments and suggestions to improve the paper.

Queries are given in black and answers in blue.

  1. What are the advantages and disadvantages of current materials other than nanocomposites? In comparison, what are the advantages of nanocomposites?

A: Thanks for your question. Conventional materials generally present diverse limitations, such as low specific surface area, high bandgap energy, instability to pH, and low ability to regenerate and reuse. Considering nanoscale phenomena, such as surface and quantum confinement effects, nanomaterials demonstrate diverse behavior about the micrometer scale and consequently exhibit new properties and characteristics. Nanocomposites are formed by two or more materials aiming to overcome their drawbacks. Therefore, these compounds showed more resistance against acid erosion, higher surface area, and performance as adsorbents and photocatalysts. In addition, the magnetic properties enhance the separation from the aqueous medium, avoiding the subsequent steps.

  1. In this study, why are conventional nanocomposites and magnetic nanocomposites divided into two categories? Is it possible to divide them according to other criteria, such as photocatalytic nanomaterials?

A: Thank you for the observation. Nanocomposites can be synthesized using different materials (reinforcement), i.e., they do not necessarily exhibit magnetic behavior. Materials were divided into two categories considering unique properties presented by magnetic nanocomposites, such as easy purification and separation from the aqueous medium. In addition, they are known to show excellent regeneration capacity, increasing the material's useful life. Supported nanocatalysts are a type of nanocomposite that aims to overcome the limitations of conventional heterogeneous catalysts, such as particle agglomeration, low dispersion in aqueous solution, and high bandgap energy.

  1. In the conclusion section, the authors should indicate the future directions.

A: Future directions and perspectives were added to the conclusion section. (See main manuscript, lines 581-591).

----------------------------------------------

Round 2

Reviewer 1 Report

The authors have made the proposed changes.

I recommend publishing the article.

Author Response

Thank you for the positive evaluation